# PhyMAGIC: Physical Motion-Aware Generative Inference with Confidence-guided LLM

## Abstract

Recent advances in 3D content generation have amplified demand for dynamic models that are both visually realistic and physically consistent. However, state-of-the-art video diffusion models frequently produce implausible results such as momentum violations and object interpenetrations. Existing physics-aware approaches often rely on task-specific fine-tuning or supervised data, which limits their scalability and applicability. To address the challenge, we present PhyMAGIC, a training-free framework that generates physically consistent motion from a single image. PhyMAGIC integrates a pre-trained image-to-video diffusion model, confidence-guided reasoning via large language models (LLMs), and a differentiable physics simulator to produce 3D assets ready for downstream physical simulation without fine-tuning or manual supervision. By iteratively refining motion prompts using LLM-derived confidence scores and leveraging simulation feedback, PhyMAGIC steers generation toward physically consistent dynamics. Comprehensive experiments demonstrate that PhyMAGIC outperforms state-of-the-art video generators and physics-aware baselines, enhancing physical property inference and motion–text alignment while maintaining visual fidelity.

## 1 Introduction

Realistic 3D content generation is essential for immersive gaming, robotics simulation, and digital twins. Recent advances in geometry and texture synthesis (Xiang et al., 2025; Voleti et al., 2024; Chen et al., 2024b; Tang et al., 2024; Miao et al., 2025) have significantly improved the fidelity of static 3D models. Extending these successes to *dynamic* content, however, remains challenging. Ensuring that motion remains physically consistent is particularly difficult. Current diffusion-based video generation models (Yang et al., 2025; Peng et al., 2025) emphasize perceptual realism but frequently violate fundamental physical principles. The resulting artifacts include momentum non-conservation, object interpenetration, and unrealistic material responses, which degrade user immersion and limit deployment in safety-critical domains such as robotics and physics-based simulation.

The challenges motivated research on *physics-aware generative methods* (Meng et al., 2025; Lin et al., 2025a). Most approaches follow one of two paradigms: (1) embedding physical priors directly into network architectures (Xu et al., 2024; Cao et al., 2024), which can improve physical consistency but often reduces generalization to novel scenes and motions, and (2) vision-based refinement methods (Liu et al., 2024b; Tan et al., 2024; Liu et al., 2025) that optimize outputs from visual observations but depend on large annotated datasets and are vulnerable to perceptual ambiguities. These methods represent an essential step toward physics-aware generation, but their effectiveness can depend strongly on the availability and quality of visual evidence.

This sensitivity becomes especially pronounced when inferring intrinsic physical properties, such as mass, density, and elasticity, from minimal visual input like a single static image. In such under-constrained settings, prior knowledge or learned estimators become critical for resolving ambiguities, yet existing solutions remain limited. Manually specified priors such as PhysGaussian (Xie et al., 2024) lack adaptability, while learning-based estimators (Cai et al., 2024; Zhang et al., 2024; Lin et al., 2025b) are susceptible to data scarcity and domain shift. Recent LLM-based methods (Liu et al., 2024b; 2025) provide a promising direction for incorporating high-level reasoning, but act less reliably with sparse visual cues and are rarely coupled with generative models in a closed-loop fashion. These limitations underscore the need for a framework that unifies reasoning and generation,

Figure 1: Static input image and generated motion sequences with the guidance of different motion types can yield distinct physical properties. For a static car input, with three motions ("free fall", "bomb", and "squeeze") guidance, the output videos are suitable for distinct physics reasoning. Video generation and physical reasoning models in this setting are CogVideoX (Yang et al., 2025) and GPT-4o (Achiam et al., 2023).

enabling the reliable inference of physical properties from a single image without requiring costly retraining or manual annotation.

To address this gap, we propose **PhyMAGIC**, a framework that operates without additional training for visual-to-physical inference and motion generation from a single image. Our key insight is that combining the reasoning ability of LLMs with the motion diversity of pretrained video diffusion models can resolve the ambiguity of single-image inference. As illustrated in Figure 1, different motion types reveal distinct levels of physical evidence: for example, a "squeeze" motion of a toy car yields more accurate mass inference than a "bomb" motion. This observation underscores that different motion trajectories reveal different levels of physical evidence. Inspired by this, PhyMAGIC synthesizes diverse motion videos from the input image, evaluates confidence in inferred physical properties such as material, mass, or elasticity, and then uses differentiable simulation to guide the next round of motion generation.

Guided by this observation, PhyMAGIC executes a three-stage, closed-loop process that actively couples generation, reasoning, and simulation. First, it *unlocks* diverse motion hypotheses by synthesizing motion-rich video candidates from the input image using a pretrained image-to-video diffusion model (Yang et al., 2025), transforming a single frame into a rich source of temporal evidence. Second, it *strategically refines* these motions through an iterative loop, where LLM-derived confidence scores (Achiam et al., 2023) guide the regeneration of targeted motion contexts that maximally reduce uncertainty in physical property estimates. Finally, it *closes the loop* by verifying and correcting the generated dynamics via a differentiable Material Point Method (MPM) simulator (Jiang et al., 2015), initialized from 3D Gaussian reconstructions (Xiang et al., 2025), enabling near-interactive, mesh-free validation of physical plausibility. Together, these stages form a self-reinforcing reasoning–generation cycle that progressively improves physical plausibility, without any annotated data or model fine-tuning.

Our main contributions are summarized as follows:

- **Efficient single-image physical inference.** We present, to the best of our knowledge, one of the first frameworks that integrates pretrained video diffusion models with iterative LLM feedback to infer mass, density, and elasticity from a single image, requiring minimal annotation and no task-specific fine-tuning.
- **Confidence-guided motion refinement.** We introduce an iterative feedback mechanism where LLM-derived confidence scores guide prompt updates, synthesizing motion contexts that maximize uncertainty reduction and improve inference precision.
- **Differentiable simulation integration.** We combine inference with an MPM-based simulator operating on 3D Gaussian reconstructions, providing mesh-free, real-time validation of physical consistency and improving robustness.
- **Comprehensive evaluation.** Experiments on PhysGaussian and Internet-collected scenes show that PhyMAGIC can achieve over 88.95% physical property inference accuracy of ground truth with the guidance of LLM-based reasoning. PhyMAGIC surpasses the strongest physics-aware baseline $10\times$ in time-consuming and 16.1% in semantic similarity. Our method also achieves strong stability and physical plausibility in dynamics generation compared with cross-category video generators, while preserving visual fidelity.

## 2 RELATED WORKS

**3D Dynamic Generation.** Benefiting from advances in static 3D representation methods, recent dynamic 3D generation approaches (Li et al., 2024; Ren et al., 2024) extend static representations into temporal sequences by explicitly modeling object trajectories or applying deformation fields. Neural implicit-based methods (Du et al., 2021; Pumarola et al., 2021) integrate spatial coordinates, timestamps, and viewing directions through learned networks to achieve photorealistic novel-view synthesis in dynamic scenarios. However, these methods generally require high computational resources due to their reliance on voxel-based rendering. Recent developments in 3D Gaussian Splatting (3DGS) (Kerbl et al., 2023) enable real-time rendering, with extensions (Katsumata et al., 2024) capable of limited geometric deformations. Nevertheless, these approaches often overlook explicit physical modeling, yielding visually plausible but physically implausible dynamics.

**Physics-Grounded Generative Models.** Ensuring physically consistent dynamic generation has prompted explorations into physics-grounded generative models. Physically embedded generative networks (Xu et al., 2024; Cao et al., 2024) explicitly integrate physical constraints, such as elasticity, density, and collision responses, into network architectures to produce physically plausible outputs. However, these approaches typically involve task-specific or material-dependent designs, which can potentially restrict their generalization capabilities. Alternatively, vision-based refinement frameworks (Liu et al., 2024b; 2025; Tan et al., 2024) decouple visual generation from physical reasoning by first synthesizing dynamics using standard generative models and subsequently refining them via physics-based objectives. While offering flexibility across various scenarios, these methods may encounter optimization challenges, limited physical supervision during the initial generation phase, and difficulties in precisely controlling detailed physical attributes.

**Large Language Model Reasoning.** Inferring physical attributes from sparse visual inputs remains challenging for physics-aware modeling. Traditional approaches (Xie et al., 2024; Tan et al., 2024; Zhang et al., 2024) often rely on supervised regression using extensively annotated video datasets, which require significant annotation effort and often struggle to generalize to unseen materials or environments. Recently, large language models (Achiam et al., 2023) have demonstrated promise for physical reasoning tasks by leveraging implicit knowledge of materials and motion behaviors. Methods such as PhysGen (Liu et al., 2024b) and PhysFlow (Liu et al., 2025) infer mass, elasticity, and force directions from static images, yet their accuracy can degrade when temporal cues are missing. Augmenting single-image inputs with auxiliary synthetic video generation (Yang et al., 2025) enriches motion context but does not guarantee physically coherent dynamics, which may affect inference reliability. Taken together, these limitations motivate approaches that iteratively use LLM reasoning to guide the synthesis of physically meaningful motions. Incorporating simulation feedback can further refine the generated dynamics, enabling the reliable inference of physical properties from minimal visual cues without the need for costly retraining or extensive annotation.

## 3 PRELIMINARY

In this section, we introduce the mathematical foundations and notation underlying our framework, with a focus on Gaussian Splatting for 3D representation and the Material Point Method for differentiable physical simulation in 3D space.

**3D Gaussian Splatting (3DGS).** 3D Gaussian Splatting (Kerbl et al., 2023) represents a scene as a collection of anisotropic Gaussian kernels $\{x_k, \Sigma_k, \alpha_k, c_k\}, k \in \mathcal{K}$, encoding position, covariance matrix, opacity, and view-dependent color. Unlike implicit neural representations (Mildenhall et al., 2020; Zhang et al., 2021), 3DGS features a fully differentiable rasterization rendering process, where the pixel color $C(u)$ is computed by alpha blending depth-sorted Gaussians:

$$C(u) = \sum_{k \in \mathcal{P}(u)} \alpha_k \, \mathcal{G}_k(u) \, \mathrm{SH}(d_k; c_k) \prod_{j<k} (1 - \alpha_j \, \mathcal{G}_j(u)). \tag{1}$$

Here $\mathcal{G}_k(u)$ is the current Gaussian projection at pixel $u$, and SH denotes spherical harmonics for view-dependent colors $c_k$ of viewing direction $d_k$. Crucially, 3DGS exposes per-Gaussian parameters that are directly optimizable, which we later exploit to reconstruct and simulate deformable objects from single images.

**Continuum Mechanics and Material Point Method.** We model material behaviors using continuum mechanics (Jiang et al., 2015; Reddy, 2013) that describes local deformation by a deformation

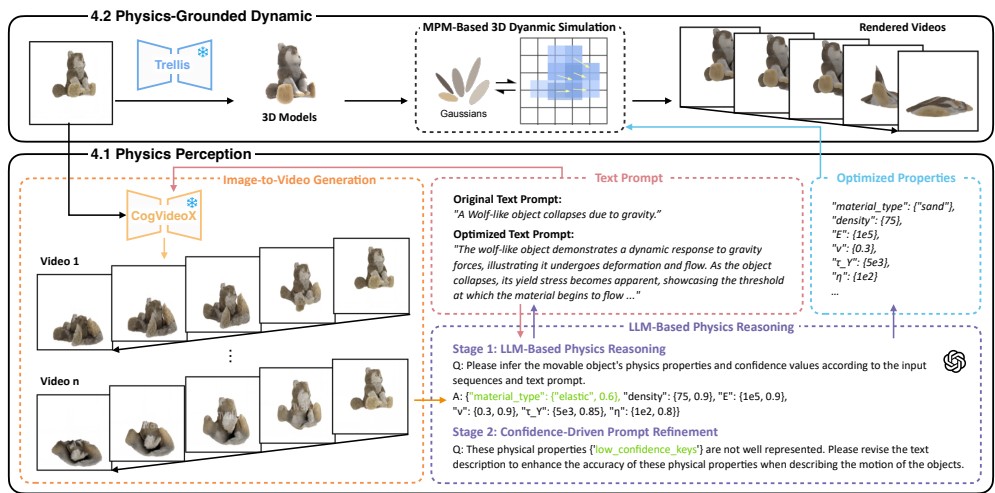

Figure 2: Overview framework of our method. PhyMAGIC consists of two stages: Physics Perception and Physics-Grounded Dynamic. The Physics Perception Stage utilizes pre-trained video generation models and LLM to obtain optimized physical properties. Physics-Grounded Dynamic Stage combines optimized physical properties into 3D models with the MPM simulator to generate rendered videos.

map $\phi(\mathbf{X}, t)$, relating initial coordinates $\mathbf{X}$ to current positions $\mathbf{x} = \phi(\mathbf{X}, t)$. For numerical simulation, we employ the Material Point Method (MPM) (Jiang et al., 2015; Zong et al., 2023), a hybrid Lagrangian–Eulerian approach that discretizes materials into particles $\{\mathbf{x}_p, m_p, \mathbf{v}_p, \mathbf{F}_p\}_{p=1}^{N}$ carrying mass, velocity, and deformation state, while using a background grid to compute forces and spatial derivatives. In MPM, each simulation step alternates between: (1) particle-to-grid (P2G) transfer of mass and momentum, (2) grid update by solving discretized momentum equations under internal and external forces, and (3) grid-to-particle (G2P) transfer to update particle velocities, positions, and deformation gradients. This formulation naturally handles large deformations, collisions, and multi-material interactions with numerical stability, and its differentiable implementation allows seamless integration into our closed-loop generation framework.

# 4 METHODOLOGY

We propose **PhyMAGIC**, a training-free framework for inferring physical properties and simulating physically plausible 3D dynamics from a single image. As illustrated in Figure 2, PhyMAGIC consists of two iteratively coupled stages: (1) a *physics perception stage*, which infers object-level physical parameters, such as Material type, Density, and Young's modulus within an LLM-guided iterative video reasoning loop, and (2) a *physics-grounded dynamic stage*, which simulates time-varying 3D motion using a differentiable MPM solver operating on Gaussians. By integrating video generation, confidence-guided prompt refinement, and simulation feedback, PhyMAGIC forms a closed-loop system that progressively aligns synthesized motion with fundamental physical principles such as momentum conservation and collision-free interactions. This closed-loop design improves the reliability of property inference and enhances the fidelity of the generated dynamics.

## 4.1 PHYSICS PERCEPTION STAGE

The physics perception stage infers object-level physical properties from a single static image by integrating visual dynamics with structured LLM reasoning. It consists of three components: (1) image-to-video generation to enrich visual cues, (2) LLM-based physics reasoning to estimate physical parameters in a model-agnostic, annotation-free manner, and (3) confidence-driven prompt refinement to iteratively resolve low-confidence predictions.

**Image-to-Video Generation.** Inferring physical properties from a single frame is fundamentally under-constrained, as attributes such as mass, elasticity, or external forces often become apparent only through temporal behavior. To enrich these cues, we generate motion-rich video sequences $V_0 = \{I_0, I_1, \ldots, I_T\}$ from the input image $I_0$ using CogVideoX1.5-5B (Yang et al., 2025), a state-of-the-art transformer-based diffusion model known for strong temporal coherence and motion diversity. To avoid redundancy and highlight meaningful motion, we apply motion-aware subsampling based on optical flow magnitude, keeping frames with significant motion changes. This transforms a single static observation into a temporally dense sequence where cues such as free-fall trajectories,

collisions, or deformations become visible, providing the subsequent LLM reasoning module with richer evidence for robust property estimation.

**LLM-Based Physics Reasoning.** Although synthesized videos reveal richer temporal cues than single images, directly inferring physical properties remains challenging due to sparse supervision and the inherent complexity of physics-aware reasoning. Pretrained LLMs can leverage implicit physical knowledge but are not explicitly trained for this task, which can lead to ambiguous or inconsistent predictions. To address these limitations, we design a structured, coarse-to-fine inference framework using GPT-4o (Achiam et al., 2023), which combines synthesized videos with targeted descriptive prompts to enhance robustness and interpretability.

As illustrated in Stage 1 of Figure 2, our reasoning process proceeds in three stages. First, the LLM identifies the primary movable object in the generated video $V_0$, isolating it from background clutter and static distractors. Next, it assigns the object a material category by combining visual appearance cues (e.g., texture, deformation) with the textual context provided in the prompt. Finally, conditioned on the material type, the LLM estimates both static properties $P_s$ (mass, density, elasticity) and dynamic parameters $P_d$ (external forces, initial velocity). Each inferred attribute is accompanied by a confidence score $c_i \in [0, 1]$, computed through a self-consistency check over multiple sampled responses (see Supplementary for details). Attributes with $c_i < \gamma$ are marked as uncertain and targeted for refinement. This explicit confidence evaluation enables the next stage to adaptively generate motion that resolves ambiguity, forming the core of our closed-loop perception pipeline.

**Confidence-Driven Prompt Refinement.** We leverage the confidence scores from the previous stage to adaptively refine textual prompts, so that the next round of synthesized motions provides more discriminative physical evidence for low-certainty attributes. However, direct LLM predictions still suffer from prompt–motion misalignment or insufficient motion cues, which may lead to physically inaccurate or ambiguous estimations. As illustrated in Figure 3, a carnation is mislabeled as a non-Newtonian fluid due to vague textual descriptions and insufficient motion evidence.

To systematically address this limitation, we introduce a confidence-guided refinement loop. We first define a confidence threshold $\gamma$ and identify attributes $P_i$ whose confidence scores fall below this threshold. These attributes are marked using a binary indicator:

$$m_i = \begin{cases} 1, & c_i < \gamma, \\ 0, & c_i \geq \gamma, \end{cases} \quad (2)$$

where all indices $i$ with $m_i = 1$ form the set of attributes requiring refinement. Starting from the initial descriptive prompt $p^{(0)}$, we update it iteratively at step $t$ as

$$p^{(t+1)} = \textbf{LLM}(p^{(t)}, \{P_i | m_i = 1\}), \quad (3)$$

where the LLM receives the current prompt and low-confidence attributes, then generates a new prompt that explicitly requests more informative motion patterns for those attributes.

Having identified low-confidence attributes through $c_i < \gamma$ [1], we iteratively refine the generation process to elicit motion evidence that resolves these ambiguities. At each iteration, we update the descriptive prompt, regenerate a new video sequence, and recompute confidence scores until all attributes exceed the threshold $\gamma$. This process adaptively requests informative motion patterns such as free-fall or squeezing actions that reveal key physical cues and progressively improve inference reliability. The resulting high-confidence attributes and motion-enriched videos provide a structured description of the scene's physical state. These outputs initialize the physics-grounded dynamic stage, where volumetric reconstruction and differentiable simulation are performed.

## 4.2 PHYSICS-GROUNDED DYNAMIC STAGE

The outputs of the physics perception stage provide both object-level physical attributes and motion-enriched videos, which serve as priors for physically realistic simulation. While Section 4.1 focuses on identifying *what* objects are present and *what* their physical properties are, this stage addresses *how* these objects move and interact over time. We explicitly bridge the gap between inferred physics

---

[1] The threshold $\gamma = 0.6$ is determined through the LLM multiple reasoning results. We find that the general inference confidence value of GPT-4 is around 0.7. Setting $\gamma$ to 0.6 can effectively focus on a few low-confidence parameters.

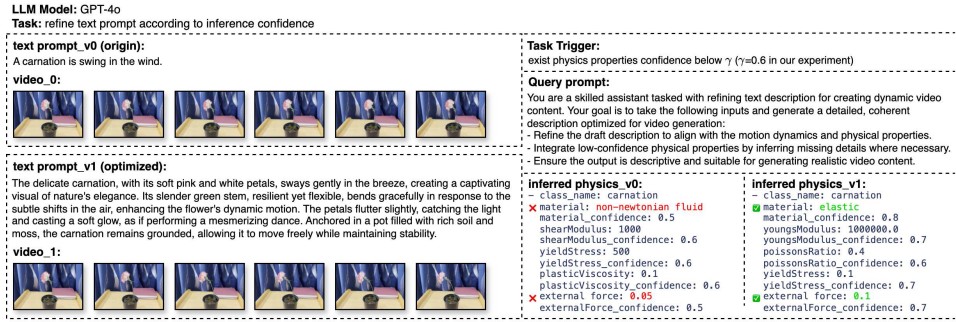

Figure 3: Confidence-driven text prompt refinement process. The presented *swing carnation* scene shows the comparison of an initial and optimized text prompt and physics properties reasoning of the two phases.

and volumetric simulation by reconstructing a 3D representation from the input image and embedding the inferred attributes into a differentiable MPM solver. This enables controllable, physically coherent, and temporally stable dynamics from a single image.

**Single Image to 3D Generation.** Generating dynamics from a single image remains a challenging task, particularly for scenarios involving non-planar motion and complex object–material interactions (Liu et al., 2024b). Conventional image-level simulation approaches (Sanchez-Gonzalez et al., 2020; Chen et al., 2024a) approximate dynamics using image cues but typically treat material properties implicitly. This may reduce accuracy in cases requiring fine-grained modeling of deformable objects or heterogeneous materials. To overcome these limitations, we reconstruct a volumetric 3D representation using a pretrained 3D Gaussian Splatting (3DGS) model (Xiang et al., 2025), which produces a set of structured Gaussian kernels

$$G_k = (x_k, \Sigma_k, \alpha_k, c_k)$$

representing position, covariance, opacity, and color. Each Gaussian kernel serves as a material-carrying particle, allowing inferred attributes such as mass and elasticity to be directly injected into the simulation process.

**MPM-Based 3D Dynamic Simulation.** Existing differentiable simulators (Abou-Chakra et al., 2024; Le Cleac'h et al., 2023; Zhong et al., 2024) often assume predefined material parameters and are not designed to integrate externally inferred attributes. Our approach explicitly couples the inferred physical properties with a differentiable MPM solver, significantly improving realism and controllability. We adopt a hybrid Lagrangian–Eulerian MPM formulation (Jiang et al., 2015) capable of robustly modeling diverse material behaviors, including granular media, elastic solids, sands, and metals. Crucially, rather than manually specifying material parameters, we initialize particle states directly from the Gaussian representation and inferred attributes:

$$\mathcal{G}_p^t = (x_p^t, \Sigma_p^t, \alpha_p, c_p, \theta_p^t), \tag{4}$$

where $\theta_p^t$ encodes properties such as mass, density, and elasticity. At each time step $t$, particle positions, velocities, and deformation gradients are updated according to MPM dynamics:

$$v_p^{t+1} = v_i^{t+1}, \quad x_p^{t+1} = x_p^t + \Delta t v_p^{t+1}, \quad \mathbf{F}_p^{t+1} = (\mathbf{I} + \Delta t \nabla v_p)\mathbf{F}_p^t, \tag{5}$$

where $v_i^{t+1}$ is the grid velocity, $\nabla v_p$ is the velocity gradient ensuring local deformation consistency, and $\mathbf{I}$ denotes the undeformed state. This process produces physically coherent trajectories that respect conservation laws and material-specific behavior.

By explicitly coupling the LLM-inferred physical attributes with a differentiable MPM solver, PhyMAGIC transforms abstract reasoning outputs into actionable particle-level dynamics. This tight integration ensures that the synthesized motions are not only visually plausible but also physically verifiable, providing controllable trajectories that directly build on the results of the perception stage.

## 5 EXPERIMENTS

We evaluate PhyMAGIC in a training-free setting by integrating pretrained Trellis (Xiang et al., 2025) for 3D Gaussian reconstruction, CogVideoX-5b (Yang et al., 2025) for video generation (720p, 50 frames), GPT-4o (Achiam et al., 2023) for physical reasoning, and a differentiable MPM solver (Xie et al., 2024) implemented in Warp (Macklin, 2022). Simulations utilize 128–200 steps,

Table 1: Quantitative evaluation on CLIP similarity and Aesthetic score with video generation methods across eight scenarios. Best results in each scenario are in **bold**, and the second-best results are in underline.

| Method | Metrics | Swing ficus | Sand wolf | Driving car | Rolling basketball | Tear toast | Sway tree | Lifting hat | Swing carnation | Average |
|---|---|---|---|---|---|---|---|---|---|---|
| OpenSora2.0 | CLIP$_{sim}$ ↑ | 0.270 | **0.264** | 0.195 | 0.230 | 0.274 | 0.174 | 0.239 | 0.217 | 0.233 |
| | Aesthetic ↑ | 27.26 | 14.58 | 5.51 | 17.74 | 23.85 | 17.44 | 2.53 | 26.97 | 16.98 |
| CogVideoX | CLIP$_{sim}$ ↑ | 0.279 | 0.225 | 0.241 | 0.226 | 0.250 | 0.186 | 0.252 | 0.254 | 0.239 |
| | Aesthetic ↑ | 33.19 | 22.55 | 38.55 | 11.91 | 47.18 | 21.51 | 16.37 | **54.02** | 30.66 |
| CogVideoX* | CLIP$_{sim}$ ↑ | 0.257 | 0.263 | 0.234 | 0.233 | **0.279** | 0.195 | 0.241 | 0.254 | 0.240 |
| | Aesthetic ↑ | **34.89** | **23.94** | **41.20** | 17.95 | 46.49 | 22.53 | 16.41 | 51.47 | **31.86** |
| **Ours** | CLIP$_{sim}$ ↑ | **0.294** | 0.200 | **0.252** | **0.266** | 0.231 | **0.243** | **0.270** | **0.256** | **0.251** |
| | Aesthetic ↑ | 31.00 | 18.38 | 29.40 | **26.82** | **49.85** | **39.67** | **26.74** | 23.70 | 30.69 |

Table 2: Quantitative results on CLIP similarity with physics-aware generation methods across six representative scenes. Best results are highlighted in **bold**.

| Method | Swing ficus | Sand wolf | Lifting hat | Sway tree | Swing carnation | Rolling basketball | Average |
|---|---|---|---|---|---|---|---|
| OMNIPHYSGS | 0.227 | 0.167 | - | - | - | - | 0.197 |
| PhysDreamer | 0.223 | 0.145 | 0.239 | 0.186 | **0.272** | - | 0.213 |
| Physics3D | 0.225 | 0.147 | 0.229 | 0.147 | 0.269 | **0.283** | 0.217 |
| **Ours** | **0.294** | **0.200** | **0.270** | 0.226 | 0.256 | 0.266 | **0.252** |

Table 3: Quantitative results on Image and Motion-FID of generated videos with the GT input images.

| Method | Image-Motion-FID↓ |
|---|---|
| PhysDreamer | 107.4652 |
| OMNIPHYSGS | 106.5992 |
| Physics3D | 98.9148 |
| **Ours** | **94.6884** |

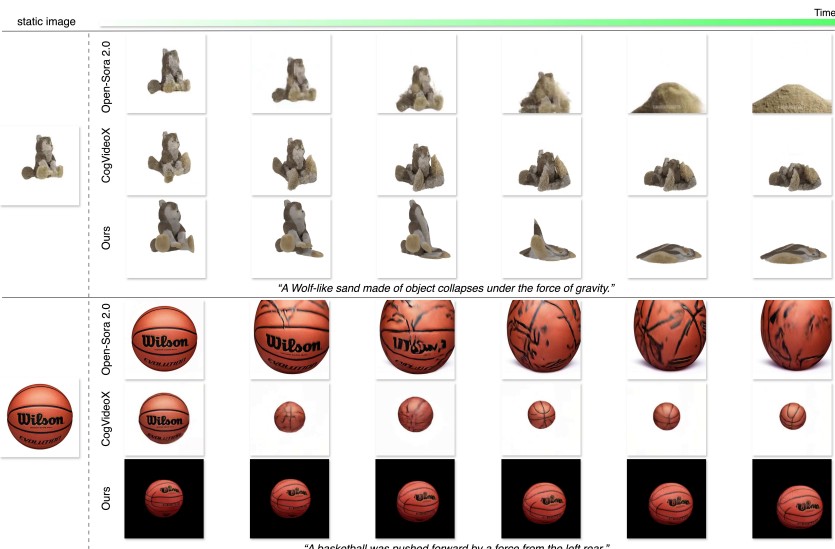

Figure 4: Qualitative comparisons of dynamic scene generation between our method and state-of-the-art video generation models (Peng et al., 2025; Yang et al., 2025). Given only a single static image (leftmost), our approach effectively infers intrinsic physical properties. It generates highly realistic dynamics over time (left to right), demonstrating superior physical realism and temporal consistency.

depending on scenario complexity, and run on a single NVIDIA RTX 4090 GPU. We benchmark on PhysGaussian (Xie et al., 2024), PhysGen (Liu et al., 2024b), and Internet-collected single-image scenes. Evaluation metrics include CLIP similarity for text–motion alignment, aesthetic score for visual quality, Image-Motion-FID for evaluating the distribution of ground-truth images with generated videos, and a 61-participant user study for physical plausibility and text consistency. We compare with state-of-the-art video generation models (CogVideoX, Open-Sora 2.0) and physics-aware approaches (PhysDreamer, Physics3D, OMNIPHYSGS). CogVideoX* denotes CogVideoX augmented with our confidence-driven prompt refinement for ablation analysis. Additional implementation settings, video generator choice, and user study details are in the Appendix A.1.

## 5.1 QUANTITATIVE RESULTS

**Comparison with image-to-video generation models.** Leveraging iterative LLM-guided reasoning and a physically grounded MPM simulator, PhyMAGIC generates physically plausible 3D dynamics from only a single input image and text prompt. Table 1 reports quantitative comparisons with state-of-the-art image-to-video generation models. Our method achieves the highest CLIP similarity scores in most scenarios, indicating stronger semantic alignment between generated motions and

Table 4: Human evaluation on physical plausibility and text consistency across eight real-world scenes compared to three image-to-video generation models. Our method achieves the highest human scores in both terms.

| Method | Physical Plausibility↑ | Text Consistency↑ |
|---|---|---|
| OpenSora2.0 | 2.02 | 2.18 |
| CogVideoX | 2.58 | 2.49 |
| CogVideoX[*] | 2.97 | 3.00 |
| **Ours** | **3.00** | **3.07** |

Table 5: Quantitative results of our PhyMAGIC and trained methods on inference speed and computational cost. We report 48 frames in all to ensure fairness. Our method achieves over 10× speedup compared with the fastest baseline, while requiring the least GPU memory.

| Methods | Physics + Simulation (s) | Relative Speedup | Device (GB) |
|---|---|---|---|
| OMNIPHYSGS (trained) | ∼8e4 + 23.83 | 1× | 14 |
| PhysDreamer (trained) | ∼5e4 + 152.02 | ∼1.60× | 40 |
| GIC (trained) | ∼1.5e4 + 138.68 | ∼5.29× | 24 |
| **Ours (training-free)** | **∼1.4e3 + 121.19** | **∼52.60×** | **14** |

Table 6: Representative motion scenarios including ficus swing (elastic), car driving (rigid-body), and basketball rolling (deformable). We report key governing parameters (density, modulus, and friction angle) and their accuracy across iterations, showing progressive improvement and convergence toward ground truth.

| | Swing ficus | | | | Rolling basketball | | | | Driving car | | | |
|---|---|---|---|---|---|---|---|---|---|---|---|---|
| Parameter | GT | Iter1 | Iter2 | Iter3 | GT | Iter1 | Iter2 | Iter3 | GT | Iter1 | Iter2 | Iter3 |
| Material | Elastic | Elastic | Elastic | Elastic | Elastic | Elastic | Elastic | Elastic | Rigid | Elastic | Rigid | Rigid |
| Density | 400 | 600 | 250 | 250 | 1000 | 85.71 | 200 | 600 | 1200 | 1000 | 1200 | 1200 |
| Young's Modulus | 3e6 | 5e4 | 3e6 | 3e6 | 1e5 | 1e4 | 1e5 | 1e5 | 2e9 | 2.5e4 | 2e9 | 2e9 |
| Poisson's Ratio | 0.30 | 0.30 | 0.40 | 0.30 | 0.40 | 0.47 | 0.50 | 0.40 | 0.40 | 0.25 | 0.75 | 0.50 |
| Yield Stress | – | – | – | – | – | – | – | – | 6e7 | – | 3e6 | 3e7 |
| **Accuracy (%)** | – | 62.92 | 82.29 | 90.63 | – | 50.27 | 73.75 | 90.00 | – | 29.17 | 44.50 | 93.75 |

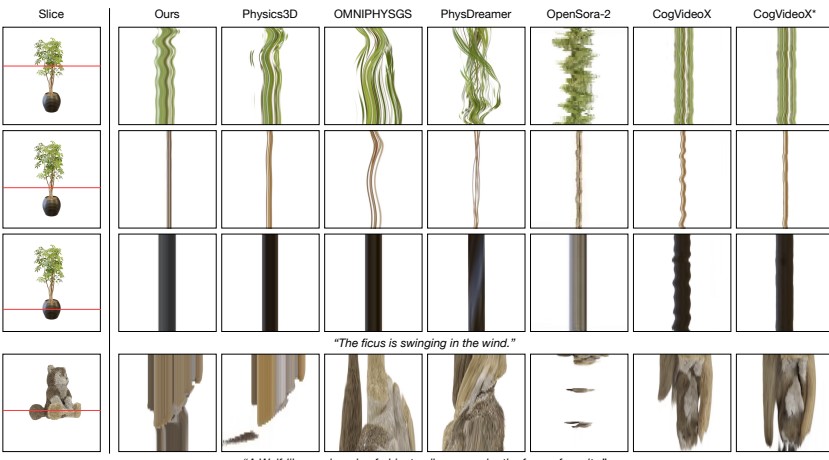

*"The ficus is swinging in the wind."*

*"A Wolf-like sand made of object collapses under the force of gravity."*

Figure 5: Space-time slices comparison with physical-aware and image-to-video generation methods.

textual descriptions. On average, PhyMAGIC reaches 0.251, outperforming OpenSora2.0 (0.233), CogVideoX (0.239), and CogVideoX[*] (0.240). Despite operating with only a single-image input, PhyMAGIC also achieves competitive performance on the Aesthetic score, surpassing other models in the *rolling basketball*, *tear toast*, *sway tree*, and *lifting hat* scenarios. These improvements stem from PhyMAGIC's explicit integration of physical priors, which guides motion synthesis toward trajectories that are not only visually appealing but also physically consistent.

**Comparison with physics-aware 3D dynamic generation models.** Table 2 compares PhyMAGIC with three representative physics-aware baselines across multiple challenging scenarios. PhyMAGIC achieves substantial gains, attaining scores of 0.294 and 0.200 on the elastic (*swing ficus*) and sand (*sand wolf*) cases, respectively. We present the distribution of the collected GT images with the generated videos in Table 3. Our method achieves the lowest Image-Motion-FID score, surpassing existing physics-aware baselines. These results highlight PhyMAGIC's ability to preserve visual fidelity while maintaining accurate text–motion coherence. Notably, PhyMAGIC improves semantic similarity by 16.1% over Physics3D, underscoring the advantage of our iterative LLM-guided refinement in producing dynamics that are both physically plausible and semantically aligned.

**Human Evaluation.** We conducted a user study to assess physical plausibility and text–motion alignment. Participants rated 32 videos on a 4-point Likert scale (1 = least, 4 = most consistent). As shown in Table 4, PhyMAGIC achieved the highest mean ratings (3.00 for plausibility, 3.07 for alignment), followed by CogVideoX[*] (2.97/3.01), which still outperformed CogVideoX and OpenSora2.0. These results confirm that confidence-guided refinement improves dynamic realism, which enables PhyMAGIC to deliver the most human-like preference motions.

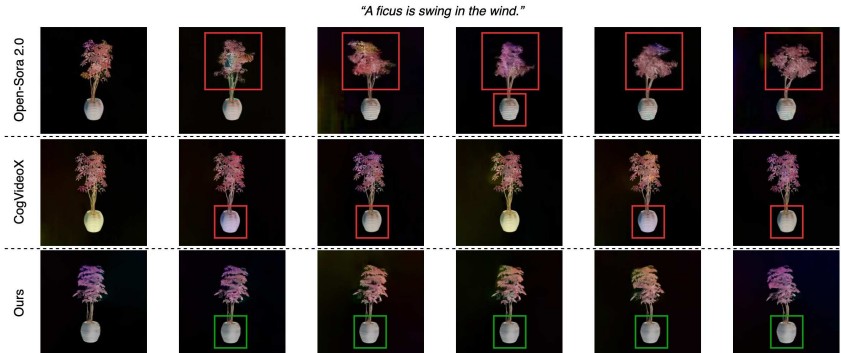

Figure 6: Qualitative comparison of optical flow visualizations on the *swing ficus* scene. PhyMAGIC maintains coherent motion and structural stability over time (green boxes), while competing methods (Peng et al., 2025; Yang et al., 2025) show structural inconsistencies and unnatural distortions (red boxes).

## 5.2 QUALITATIVE RESULTS

**Visual Comparison with Video Generation Models.** As illustrated in Figure 4, our method exhibits clear advantages in modeling physically realistic dynamics given only a static image, whereas Open-Sora 2.0 (Peng et al., 2025) and CogVideoX (Yang et al., 2025) often produce visual distortions or physically implausible motions. For the wolf-like sand collapse, Open-Sora 2.0 and CogVideoX exhibit unnatural particle scattering behaviors, whereas our results accurately simulate the behavior of sand material under gravity. In the *rolling basketball* scene, baseline models produce implausible deformations under external forces or distorted appearance, whereas our method maintains consistent appearance and motion.

**Spatiotemporal Slice Analysis.** To evaluate temporal coherence, we compare space–time slices in Figure 5. Our method produces precise and continuous motion trajectories that match expected physical laws. By contrast, Physics3D, OMNIPHYSGS, PhysDreamer, and OpenSora-2 often show blurred edges and fragmented strips, indicating weak temporal consistency and inaccurate physical dynamics. CogVideoX and CogVideoX* exhibit regular motion patterns but fail to control static regions, resulting in unintended motion in non-moving components such as the base of the ficus.

**Optical Flow Visualization.** Figure 6 visualizes optical flow (Teed & Deng, 2020) on a sample *swing ficus* scene. OpenSora2.0 exhibits spurious edges and temporal jitter around movable regions, indicating weak motion coherence, and the bottom vase also shows slight deformation. Both PhyMAGIC and CogVideoX illustrate stable optical flow; however, CogVideoX lacks precise spatial control and generates unintended motion in static regions due to its global motion modeling strategy. Our method achieves accurate motion localization through physics-aware constraints, ensuring that only the target regions exhibit dynamic deformation while static objects remain unaffected.

## 5.3 ABLATION STUDY

To assess the contribution of LLM-based physics reasoning and determine an effective number of refinement iterations, we report results in Table 6. With limited initial motion context, the first iteration often misclassifies materials (e.g., predicting a driving car as elastic) or yields inaccurate physical parameters. Subsequent iterations progressively improve parameter estimation by refining text prompts and regenerating motions. Performance stabilizes at three iterations, where the overall accuracy exceeds 90%, indicating that iterative reasoning effectively resolves early ambiguities and converges to a reliable physical understanding. More results are provided in the Appendix A.5.

## 6 CONCLUSIONS

We present PhyMAGIC, a training-free framework that achieves motion-aware generation through iterative LLM-guided video generation and differentiable MPM simulation. Our innovations lie in integrating a pretrained video generation model and a confidence-driven LLM feedback mechanism for physics reasoning optimization in the iteration. We also leverage a real-time MPM engine that simulates multi-material behaviors, guided by parameters inferred from robust physics reasoning. Extensive experimental results demonstrate that PhyMAGIC significantly improves physical plausibility in dynamic 3D assets, while maintaining high-quality text-video consistency.

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

# A  APPENDIX

## A.1  EXPERIMENTAL SETTING

**Implementation Details.** Our implementation is fully training-free, integrating several pretrained models to enable efficient and robust dynamic generation without requiring additional fine-tuning or task-specific supervision. We employ Trellis (Xiang et al., 2025) to obtain high-fidelity 3D Gaussian representations for the initial 3D reconstruction. For video generation to support effective physics inference, we use CogVideoX-5b model (Yang et al., 2025), and the generated videos are $720 \times 480$ with 50 frames. To reduce visual redundancy in subsequent LLM (Achiam et al., 2023) reasoning, we subsample 6-8 representative frames per sequence. The iteration number for confidence-driven prompt refinement is fixed to 3, ensuring a balance between inference robustness and computational efficiency. The physics-based dynamic simulation is implemented using a differentiable MPM solver (Xie et al., 2024) integrated within the Warp simulation framework (Macklin, 2022). The simulation steps across different scenarios in this experiment are configured with 128 ~200 steps. All experiments are conducted on a single NVIDIA RTX 4090 GPU with 24GB of memory.

**Benchmark.** We conduct physical reasoning and simulation experiments primarily on the real-world datasets introduced by PhysGaussian (Xie et al., 2024) and PhysGen (Liu et al., 2024b). To further demonstrate the generality of our approach, we also utilize several static images (e.g., rolling basketball and sway tree) collected from the Internet. We employ a combination of proxy metrics that can be computed from a single input image. Specifically, we report text-video CLIP similarity to compare the consistency of text-motion with physics-aware generation models and video generation models. The video-level aesthetic score is used to compute the visual aesthetics of the generated 3D assets at the rendered video level, allowing us to evaluate whether the physics-aware model significantly degrades visual appearance compared to diffusion-based video generation models.

**Video Generator Choice.** Considering the limitation of computing resources, we experimented with both OpenSora-2 and CogVideoX as candidate video generators for our framework. We found that both models yielded comparable improvements in physical accuracy, as shown in Table 7, with a difference of less than 1.3%. However, CogVideoX was substantially more resource-efficient, requiring only a single 9 GB GPU and 21 minutes for inference, whereas OpenSora-2 approximately needed 86 GB of GPU memory and 53 minutes of inference time. Based on this comparison, we selected CogVideoX as our primary video generation model due to its efficiency and open-source availability, which are key requirements for our iterative physical reasoning pipeline.

**Human Evaluation Setting.** We include human preference studies on both physical plausibility and text alignment with video generation models. We collected ratings from 61 participants on 32 videos generated by four methods. The videos are presented in a random order, and participants are asked to evaluate these videos on a 4-point Likert scale (1 = least consistent, 4 = most consistent) from both physical plausibility and textual consistency dimensions, providing a comprehensive measure of human preference.

**Baseline Models.** We compare our approach with several representative state-of-the-art methods, covering two video generation models (Yang et al., 2025; Peng et al., 2025) and three physics-aware 3D dynamic generation approaches. **CogVideoX** (Yang et al., 2025) represents a diffusion-based image-to-video generation framework that achieves stable and visually coherent motion through multi-scale temporal attention. **Open-Sora 2.0** (Peng et al., 2025) is an open-source video generation model that leverages hierarchical diffusion with spatiotemporal conditioning to synthesize high-quality sequences. We also evaluate **CogVideoX**[*] with enhancement by our confidence-driven prompt refinement module as an ablation study. For physics-aware generation baselines, we compare with PhysDreamer (Zhang et al., 2024), **Physics3D** (Liu et al., 2024a), and OMNIPHYSGS (Lin et al., 2025b). **PhysDreamer** learns physical properties from current video diffusion models, and Physics3D (Liu et al., 2024a) integrates viscoelastic MPM with Score Distillation Sampling to simulate multi-material behaviors. **OMNIPHYSGS** (Lin et al., 2025b) provides a physics-based 3D dynamic synthesis framework capable of modeling diverse material behaviors.

Table 7: Inference comparisons of OpenSora-2 and CogVideoX-5B as video generators on effect, time-consuming, and GPU needs.

| Video Generators | Inference Accuracy Effect (±%) | Inference Time (mins) | Max GPU Memory Required (GB) |
|---|---|---|---|
| OpenSora-2 | +10.9 | 53 | ∼83 |
| CogVideoX-5B | +12.2 | 21 | ∼9 |

## A.2   TEXT PROMPT FOR PHYSICS REASONING

Describe this inference without any influence from the previous conversation. Consider this a completely new interaction with no prior context.

**Step 1: Classify and determine movement**
Describe the movement object classes in the given set of images.

1. Ensure that every pixel belongs to a specific class. Use singular noun names for classes with no "-" or overlap.

2. For each class, determine if the instances are "movable" (true/false) based on:
- The object has a definite shape (e.g., car, person) and is not an amorphous area (e.g., grass, sky).
- All instances of this class are fully visible with a complete shape.
- The object's position changes over time relative to the background (for video or sequential images).

**Step 2: Determine material type**
For each object class marked as "movable" in the previous step, determine its material type based on the following observations:

1. Material Type Categories: Elastic, Plasticine, Rigid, Sand, Newtonian Fluid, Non-Newtonian Fluid.

2. Observations:
- Deformation behavior (elastic/plastic/rigid).
- Surface characteristics (gloss, texture).
- Motion behavior (interaction patterns and movement characteristics).

**Step 3: Estimate physical properties**
For each movable object with a determined material type, estimate the relevant physical properties, dynamic properties, and confidence score in $[0, 1]$. Among these, physical properties include static parameters such as mass, density, Young's Modulus, Poisson's Ratio, Yield Stress, Friction Angle, Fluid Viscosity, Bulk Modulus, and Shear Modulus. Dynamic properties contain external force $(N)$, initial velocity $(km/h)$.

1. Elastic: mass, density $(kg/m^3)$, Young's Modulus $(Pa)$, Poisson's Ratio.

2. Plasticine: mass, density, Young's Modulus, Poisson's Ratio, Yield Stress $(Pa)$.

3. Rigid: mass, density, Young's Modulus, Poisson's Ratio, Yield Stress.

4. Sand: mass, density, Friction Angle.

5. Newtonian Fluid: mass, density, Fluid Viscosity $(Pa \cdot s)$, Bulk Modulus $(Pa)$.

6. Non-Newtonian Fluid: mass, density, Shear Modulus $(Pa)$, Yield Stress $(Pa)$, Plastic Viscosity $(Pa \cdot s)$.

(The output example is on the next page.)

```
Output only the applicable properties for each material type, e.g.,
                            {
                              "class_name": "car",
                              "material": "rigid",
                              "material_confidence": 0.0,
                              "mass": 0,
                              "mass_confidence": 0.0,
                              "density": 7850,
                              "density_confidence": 0.0,
                              "youngsModulus": 2.0e11,
                              "youngsModulus_confidence": 0.0,
                              "poissonsRatio": 0.3,
                              "poissonsRatio_confidence": 0.0,
                              "yieldStress": 0.7,
                              "yieldStress_confidence": 0.0,
                              "external force": 0,
                              "externalForce_confidence": 0.0,
                              "initial velocity": 0,
                              "initialVelocity_confidence": 0.0
                            }
```

## A.3 TEXT PROMPT FOR TEXT OPTIMIZATION

> You are a skilled assistant tasked with refining text description for creating dynamic video content. Your goal is to take the following inputs and generate a detailed, coherent description optimized for video generation:
>
> 1. Origin text description.
> 2. Detected low physical properties.
>
> Your task:
> - Refine the draft description to align with the motion dynamics and physical properties.
> - Integrate low-confidence physical properties by inferring missing details where necessary.
> - Ensure the output is descriptive, coherent, and suitable for generating realistic video content.

## A.4 QUALITATIVE COMPARISON RESULTS

We conducted a series of quantitative experiments comparing our method, **PhyMAGIC**, with state-of-the-art baselines including OpenSora 2.0, CogVideoX, and CogVideoX[*] across eight diverse physical scenarios. All baselines generate videos conditioned on image and text inputs. PhyMAGIC, while also taking the same inputs, leverages a lightweight 3D reconstruction module followed by physics-based simulation using an MPM solver to model the dynamics of the foreground. While the underlying mechanisms differ slightly in representation, all methods ultimately aim to produce physically plausible motion grounded in initial visual/textual cues. This enables a fair quantitative comparison under the same input-output setting, focusing on evaluating physical consistency, generalization, and reasoning capability.

We present additional qualitative comparisons in Figure 7 and Figure 8. For fine-grained motions such as *rolling basketball*, *swinging ficus*, *swaying carnation*, and *moving car*, MAGIC consistently produces smooth, stable motion trajectories that align with physical expectations, particularly conserving momentum and maintaining directional coherence. In contrast, baseline video generation models often exhibit unstable dynamics, unnatural jitter, and physically implausible transitions. For complex object interactions like *tearing toast* or *lifting a hat*, PhyMAGIC demonstrates strong structure preservation and semantic consistency, accurately retaining the geometry and identity of input objects throughout the simulation. Optical flow (Figure 9) analysis further confirms this: in the

*swaying tree* case, PhyMAGIC captures motion focused at the crown to simulate wind-driven oscillation, while CogVideoX remains nearly static. Similarly, in the *pulling hat* scenario, PhyMAGIC restricts motion to the lower edge where the pulling force is applied, whereas baselines incorrectly induce global motion across the object. Together, these results demonstrate that PhyMAGIC not only enhances visual quality but also brings substantial improvements in physical plausibility and semantic fidelity.

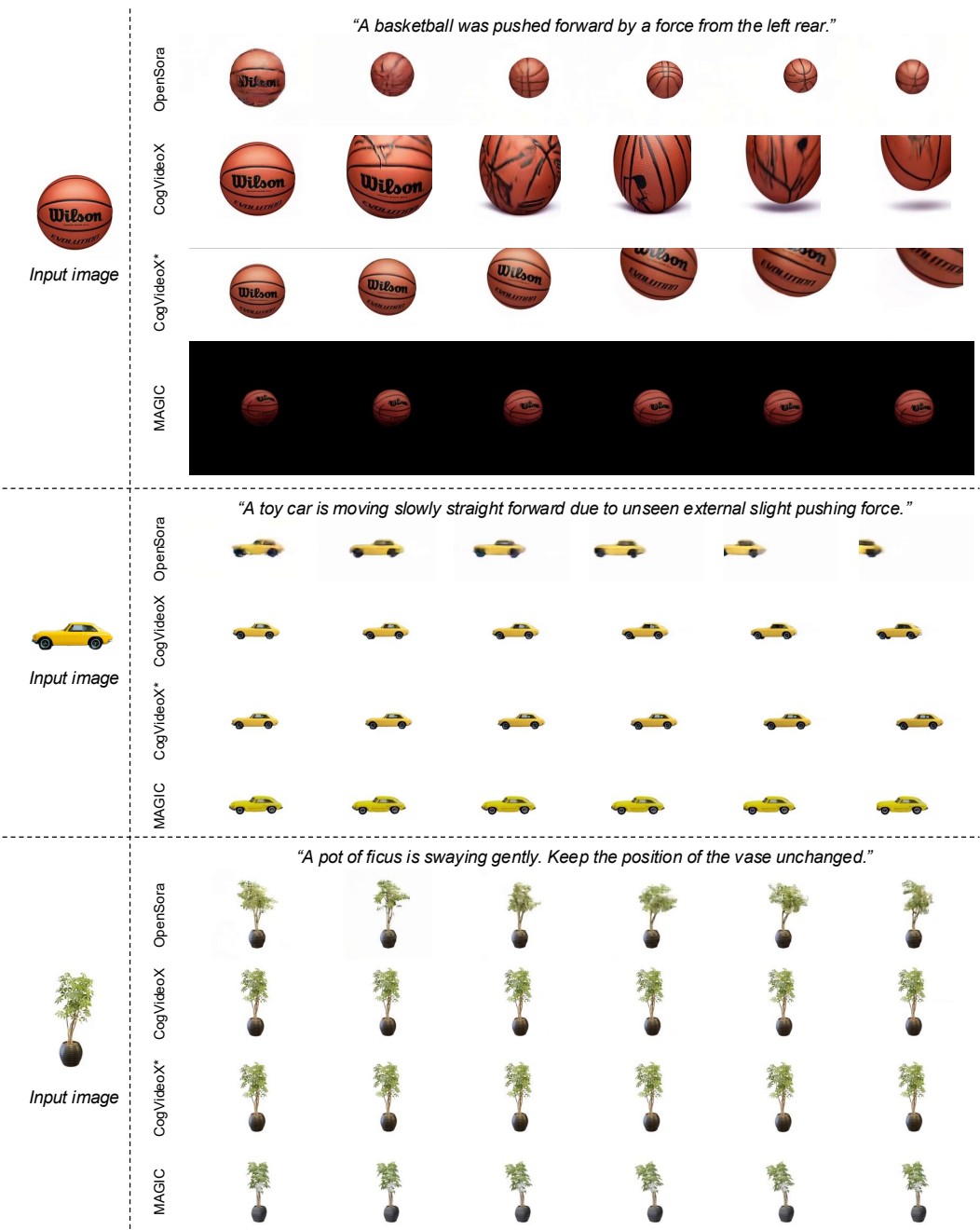

Figure 7: Qualitative comparison results with video generation models (Part 1).

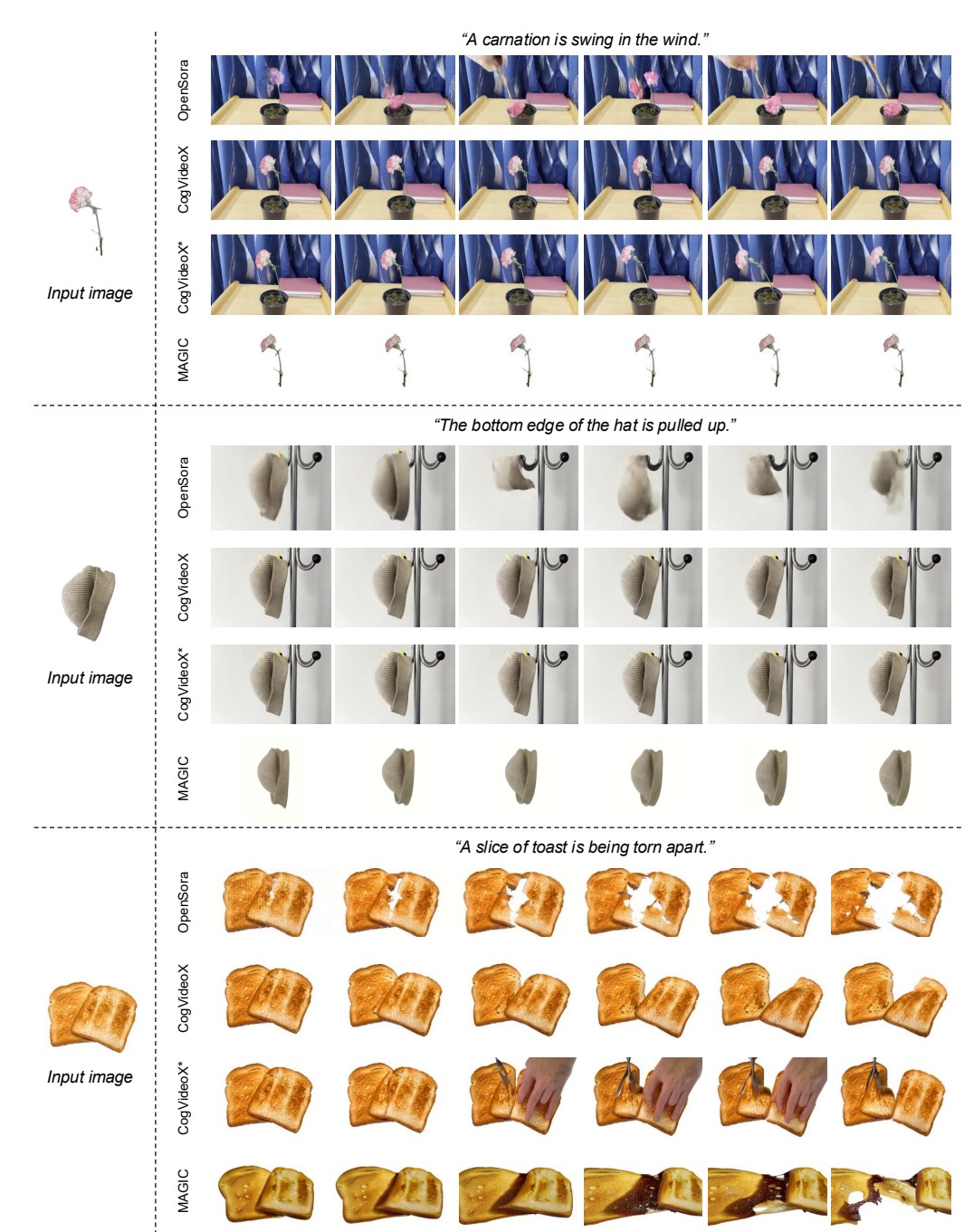

Figure 8: Qualitative comparison results with video generation models (Part 2).

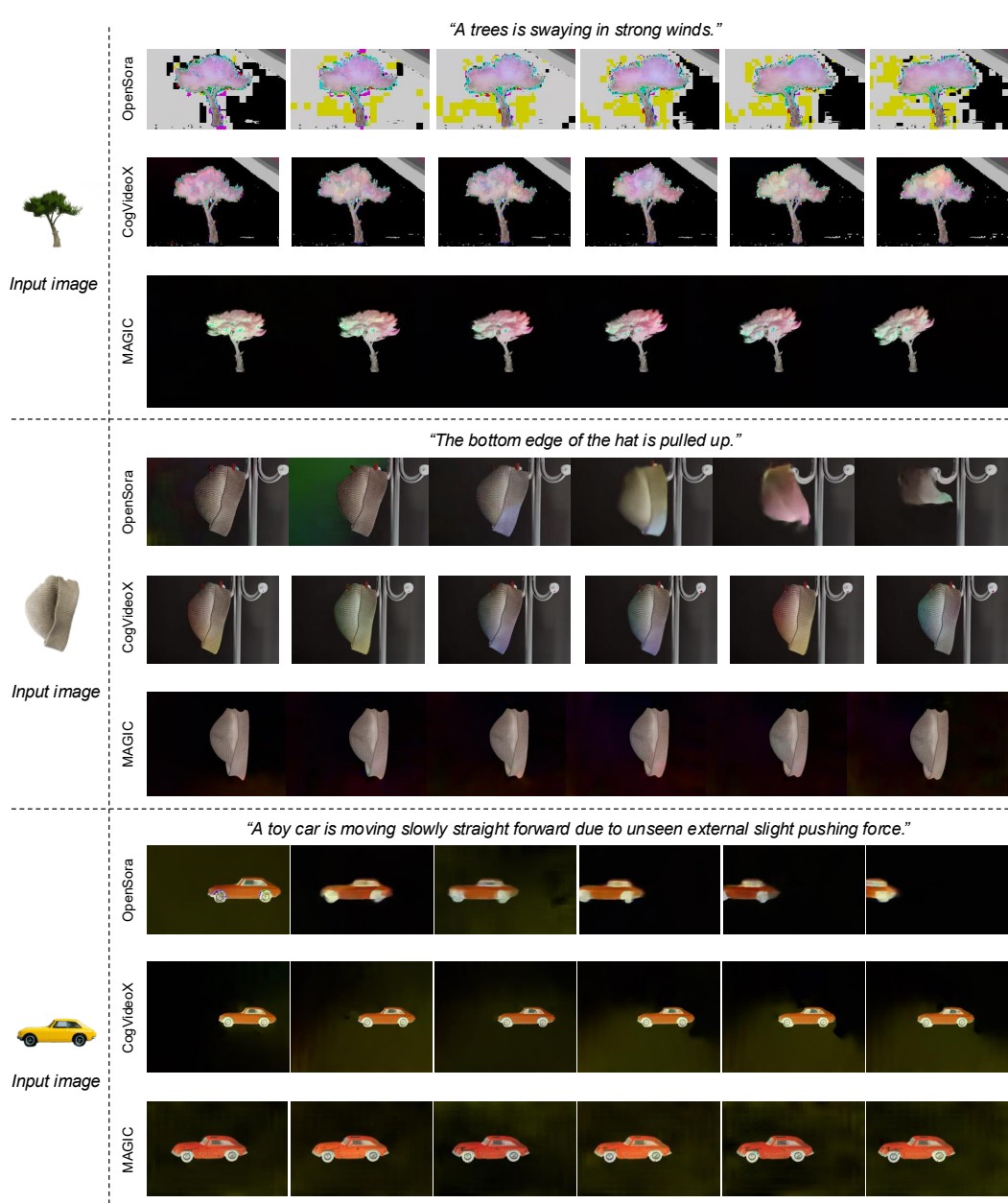

Figure 9: Optical flow visualization comparison results with video generation models.

Table 8: LLM-based physics reasoning in each iteration. For each scenario, we focus on key governing parameters such as density, modulus, and friction angle that critically determine the observed dynamics. The reported accuracy percentage quantifies how closely the inferred parameters align with the ground truth, demonstrating consistent improvement across iterations and convergence toward the correct physical properties.

| Motion Scenarios | Parameters | Ground truth | Iteration 1 | Iteration 2 | Iteration 3 |
|---|---|---|---|---|---|
| Sand wolf | Material | Sand | non-Newtonian | Sand | Sand |
| | Density | 1666 | 600 | 1666 | 1666 |
| | FrictionAngle | 40 | - | 45 | 40 |
| | ShearModulus | - | 5e4 | - | - |
| | YieldStress | - | 2e4 | - | - |
| | PlasticViscosity | - | 0.1 | - | - |
| | **Accuracy** (%) | - | **12.00** | **95.83** | **100** |
| Tear toast | Material | Elastic | Elastic | Elastic | Elastic |
| | Density | 170 | 400 | 300 | 200 |
| | YoungsModulus | 1e3 | 1e4 | 2e3 | 2e3 |
| | PoissonsRatio | 0.2 | 0.2 | 0.3 | 0.2 |
| | **Accuracy** | - | **-158.82** | **43.38** | **70.59** |
| Sway tree | Material | Elastic | Elastic | Elastic | Elastic |
| | Density | 750 | 500 | 600 | 600 |
| | YoungsModulus | 1e9 | 1e9 | 1e9 | 5e9 |
| | PoissonsRatio | 0.4 | 0.4 | 0.4 | 0.4 |
| | **Accuracy** | - | **91.67** | **95.00** | **95.00** |
| Lifting hat | Material | Elastic | Elastic | Elastic | Elastic |
| | Density | 1100 | 500 | 500 | 1250 |
| | YoungsModulus | 2e9 | 1e6 | 1e9 | 1e9 |
| | PoissonsRatio | 0.4 | 0.4 | 0.4 | 0.4 |
| | **Accuracy** | - | **61.38** | **73.86** | **84.09** |
| Swing carnation | Material | Elastic | non-Newtonian fluid | Rigid | Elastic |
| | Density | 600 | 800 | 600 | 800 |
| | YoungsModulus | 1e6 | - | 1e9 | 1e6 |
| | PoissonsRatio | 0.3 | - | 0.4 | 0.35 |
| | shearModulus | - | 1e3 | - | - |
| | YieldStress | - | 5e2 | 0.1 | - |
| | PlasticViscosity | - | 0.1 | - | - |
| | **Accuracy** | - | **16.67** | **41.67** | **87.50** |
| Swing ficus | Material | Elastic | Elastic | Elastic | Elastic |
| | Density | 400 | 600 | 250 | 250 |
| | YoungsModulus | 3e6 | 5e4 | 3e6 | 3e6 |
| | PoissonsRatio | 0.3 | 0.3 | 0.4 | 0.3 |
| | **Accuracy** (%) | - | **62.92** | **82.29** | **90.63** |
| Driving car | Material | Rigid | Elastic | Rigid | Rigid |
| | Density | 1200 | 1000 | 1200 | 1200 |
| | YoungsModulus | 2e9 | 2.5e4 | 2e10 | 2e10 |
| | PoissonsRatio | 0.40 | 0.25 | 0.75 | 0.50 |
| | YieldStress | 6e7 | - | 3e5 | 3e7 |
| | **Accuracy** (%) | - | **29.17** | **44.50** | **93.75** |
| Rolling basketball | Material | Elastic | Elastic | Elastic | Elastic |
| | Density | 1000 | 85.71 | 200 | 600 |
| | YoungsModulus | 1e5 | 1e4 | 1e6 | 1e6 |
| | PoissonsRatio | 0.4 | 0.47 | 0.5 | 0.4 |
| | **Accuracy** (%) | - | **50.27** | **73.75** | **90.00** |
| | **Average Accuracy** (%) | - | **20.66** | **68.79** | **88.95** |

## A.5 ABLATION STUDY

We evaluate the effectiveness of LLM-based iterative physical reasoning and the reasonable setting of the number of iterations in Table 8. From the experimental results, when the iteration number is set to 3, the average accuracy of all scenarios reaches its highest level, 88.95%, which is 20.16% higher than the second iteration. Considering computing resources and accuracy, this paper sets the number of iterations to 3.

### A.6 LLM Usage Statement

In the process of writing this paper, we used a large language model (ChatGPT, GPT-5, by OpenAI) to only aid in the *polishing* of the manuscript text. Specifically, LLM assistance was utilized to enhance the clarity and conciseness of the draft version and to revise sentences for grammatical correctness and improved readability.

No LLM-generated content was used without human review: all generated results were carefully checked, edited, and verified by the authors for technical correctness and consistency. No LLM was used for data generation, model training, experiment design, or result fabrication.

