# OpenReview forum: "PhyMAGIC: Physical Motion-Aware Generative Inference with Confidence-guided LLM"
_ICLR.cc/2026/Conference — ICLR 2026 Conference Withdrawn Submission_

### Official Review · Reviewer_z1t3 · 2025-10-26

**Soundness:** 2
**Presentation:** 2
**Contribution:** 2
**Rating:** 4
**Confidence:** 3

**Summary:**

The paper presents PhyMAGIC, a training-free framework that enables physically consistent 3D motion generation and physical property inference from a single static image. Unlike prior diffusion-based or physics-embedded models that require task-specific training, PhyMAGIC integrates three pretrained components—a video diffusion model (CogVideoX), a large language model (GPT-4o) for physics reasoning, and a differentiable Material Point Method (MPM) simulator—into a unified, closed-loop system. The framework iteratively refines motion generation through confidence-guided LLM feedback, where low-confidence physical attributes (e.g., density, elasticity, friction) trigger targeted prompt updates and new motion synthesis. The refined videos and inferred parameters are then validated and corrected via differentiable simulation, forming a reasoning–generation cycle that progressively improves physical plausibility without any fine-tuning or supervision.

Comprehensive experiments on benchmark datasets such as PhysGaussian, PhysGen, and various real-world single-image scenes show that PhyMAGIC achieves superior performance in both semantic and physical consistency compared to state-of-the-art video generation and physics-aware baselines. It enhances text–motion alignment (up to 16% CLIP similarity improvement) and maintains high visual fidelity while being more computationally efficient—over ten times faster than trained counterparts. Overall, the paper contributes a novel and generalizable approach that bridges LLM reasoning, diffusion-based video synthesis, and differentiable physics simulation to realize physically grounded, interpretable, and scalable dynamic generation from minimal visual input.

**Strengths:**

1. The paper introduces a conceptually elegant and practically valuable approach that unifies pretrained video diffusion models, large language model reasoning, and differentiable physics simulation into a closed-loop pipeline. This design enables physical property inference and motion synthesis from a single static image without any task-specific fine-tuning or annotated data, which is a notable advance in the direction of scalable physics-aware generation.
2. The authors provide clear physical intuition for the design choices. The observation that different motion trajectories reveal distinct levels of physical evidence (e.g., a squeeze motion discloses elasticity better than free fall) is both physically grounded and conceptually insightful, forming a solid theoretical backbone for the proposed iterative reasoning framework.
3. The explicit use of interpretable physical parameters (density, elasticity, yield stress, etc.) also makes the model’s behavior more transparent compared to end-to-end black-box video diffusion systems.

**Weaknesses:**

1. The paper does not include supplementary videos or visual demonstrations, which are crucial for evaluating a method that claims improvements in physical realism and motion plausibility. From the static figures alone, it is difficult to fully assess the perceptual quality and physical consistency of the generated dynamics. In particular, Figure 5 lacks ground-truth visualizations, making it hard to judge whether the proposed approach indeed performs better than the baselines in practice. The authors are strongly encouraged to provide additional qualitative video results for clearer evaluation.
2. Several figures are visually coarse and fail to clearly convey the intended comparisons or architectural design. For example, Figure 1 and Figure 3 appear somewhat rough and could be redrawn with more precise annotations or higher-quality visuals. Moreover, the paper does not explain clearly how the baselines are re-implemented or compared. A more transparent comparison protocol would strengthen the credibility of the experimental results.
3. The experimental section compares PhyMAGIC mainly against earlier open-source baselines such as CogVideoX, PhysDreamer, and Physics3D, but omits several state-of-the-art commercial or research-level models, including Wan 2.2, Sora, and Veo. Given the rapid progress in video generation, evaluating against these newer and higher-performing models would provide a fairer and more convincing benchmark for the claimed advances.
4. The core component of PhyMAGIC relies on the assumption that large language models can accurately infer physical parameters (e.g., density, Young’s modulus, Poisson’s ratio) from limited visual cues. However, the experimental results do not strongly support this assumption. As shown in Table 8, even after multiple refinement iterations, the predicted physical values often deviate from ground truth by one or more orders of magnitude. This raises concerns about the robustness and reliability of LLM-based physical inference.
5. While the framework leverages pretrained video diffusion models to synthesize motion evidence, current video generation models themselves often suffer from physical inconsistency, such as non-conservation of momentum and unrealistic object interactions. Relying on such imperfect priors may limit the upper bound of PhyMAGIC’s physical realism, especially when the diffusion model introduces visually plausible but physically implausible trajectories.

**Questions:**

The author did not include the code in the Supplementary Material. Do you have any plan for publishing the implementation?

---

### Official Review · Reviewer_ZtRm · 2025-10-30

**Soundness:** 3
**Presentation:** 2
**Contribution:** 2
**Rating:** 2
**Confidence:** 3

**Summary:**

The paper proposes PhyMAGIC, a training-free framework for generating physically consistent motion from a single image. It combines a pre-trained image-to-video diffusion model, LLM-based confidence reasoning, and a differentiable physics simulator to create 3D assets suitable for simulation without fine-tuning or supervision. By iteratively refining motion prompts with simulation feedback, PhyMAGIC produces realistic and physically plausible dynamics, outperforming existing video generators and physics-aware methods in both physical consistency and visual quality.

**Strengths:**

1. The writing is clear.

2. The integration of LLMs, generation models, and physics simulators is technically compelling.

3. The problem addressed is interesting and meaningful.

**Weaknesses:**

1.	What is the ultimate goal of this work — generating 3D or generating video? My understanding is that rendering videos from 3D Gaussians is the final objective, with the video primarily serving as a source of physical information. In that case, is a video generation model truly necessary? Could the physical quantities instead be provided directly by an LLM’s prior knowledge or manually specified by humans? Would this affect the final results? Overall, the connection between video generation and 3D rendering feels quite disconnected.

2.	Prior works such as DreamPhysics, PhyDreamer, and Physics3D also combine video generation with physics simulators. Could you elaborate more specifically on what the distinct contributions of this work are compared to them? Overall, I think the framework is quite similar.

3.	Since the Trellis model is used, the system can only generate single-object videos. It cannot handle scenes with backgrounds or objects of higher complexity and realism. For instance, PhysCtrl（NeurIPS25）, which also integrates a simulator with video generation, can produce more complex and natural video results.

4.	The rendered outputs suffer from the inherent granularity of Gaussians, leading to a noticeable lack of sharpness and fine details — for example, in the wolf case, most detail is lost in later frames. Moreover, as no video visualizations are provided, it is difficult to fully assess the generation quality; my observations are based solely on the few provided images.

**Questions:**

please refer to the weeknesses.

---

### Official Review · Reviewer_oTbR · 2025-11-01

**Soundness:** 3
**Presentation:** 2
**Contribution:** 2
**Rating:** 4
**Confidence:** 5

**Summary:**

This paper introduces PhyMAGIC, a training-free framework designed to generate physically realistic 3D animations from a single static image. The core problem it addresses is that modern video generation models often create visually appealing but physically implausible motion (e.g., violating momentum, objects interpenetrating).

**Strengths:**

1. The framework's strength is its training-free nature. It orchestrates existing, powerful foundation models (I2V, LLM) without requiring costly fine-tuning or massive annotated physics datasets, making it scalable and efficient.
2. The core methodological innovation is the closed-loop mechanism. The system intelligently "probes" the scene with different motions to resolve ambiguity, leading to progressively more accurate physical property inference.
3. It outperforms both standard video generation models and other physics-aware baselines in physical plausibility, as confirmed by quantitative metrics and human evaluations.

**Weaknesses:**

1. While training-free, the iterative inference process is computationally intensive and slow, requiring multiple calls to both a VDM and an LLM, followed by a physics simulation. It would be better to compare the inference time.
2.  The entire system's performance is fundamentally bottlenecked by the capabilities of its components. A failure in the VDM's ability to follow a prompt, or a flaw in the LLM's physical reasoning, will directly degrade or break the entire pipeline. More analysis of the method's limitations would be better.
3. Providing some video results will help to better observe the effectiveness of the method.
4. Some related works[1,2,3] need to be compared and discussed.

[1] PhysGen3D: Crafting a Miniature Interactive World from a Single Image. CVPR 2025.
[2] DreamPhysics: Learning Physical Properties of Dynamic 3D Gaussians with Video Diffusion Priors. AAAI 2025.
[3] GaussianProperty: Integrating Physical Properties to 3D Gaussians with LMMs. ICCV 2025.

**Questions:**

Please see Weaknesses.

---

### Official Review · Reviewer_SCqZ · 2025-11-02

**Soundness:** 3
**Presentation:** 3
**Contribution:** 3
**Rating:** 6
**Confidence:** 3

**Summary:**

PhyMAGIC is a training-free framework that generates physically consistent motion from a single image. It utilizes a closed-loop system that actively couples generation, reasoning, and simulation. The core work involves integrating a pre-trained image-to-video diffusion model and a differentiable physics simulator with confidence-guided reasoning via LLMs (GPT-4o). This process iteratively refines motion prompts using confidence scores to improve the inference of physical properties

**Strengths:**

PhyMAGIC is a training-free framework generating physically consistent motion from a single image. It uses confidence-guided LLM reasoning in a closed loop, achieving superior plausibility, motion-text alignment, and over 10× speedup compared to trained baselines.

**Weaknesses:**

The primary limitation is that inferring physical properties from a single static image is fundamentally under-constrained. This necessitates a closed-loop iterative reasoning process  to resolve ambiguities and achieve high accuracy. The framework also relies on external models, some of which require high computational resources.

**Questions:**

Did the authors conduct an ablation study to empirically confirm that γ=0.6 optimally balances inference robustness and computational efficiency?

---

### Note · Authors · 2025-12-03

I have read and agree with the venue's withdrawal policy on behalf of myself and my co-authors.